# Association between Cross-Cultural Social Adaptation and Overseas Life Satisfaction among Chinese Medical Aid Team Members (CMATMs) in Africa

**DOI:** 10.3390/ijerph16091572

**Published:** 2019-05-06

**Authors:** Xiaochang Chen, Xiaojun Liu, Wei Yu, Anran Tan, Chang Fu, Zongfu Mao

**Affiliations:** 1School of Humanities and Management, Guangdong Medical University, 1# Xincheng Road, Dongguan 523808, China; chenxiaochang@gdmu.edu.cn; 2Institute of Health Law and Policy, Guangdong Medical University, 1# Xincheng Road, Dongguan 523808, China; 3School of Health Sciences, Wuhan University, 115# Donghu Road, Wuhan 430071, China; 2016302170027@whu.edu.cn (W.Y.); chloetar@whu.edu.cn (A.T.); 4Global Health Institute, Wuhan University, 115# Donghu Road, Wuhan 430071, China; 2015103050006@whu.edu.cn

**Keywords:** cross-cultural social adaptation, overseas life satisfaction, Chinese medical aid team members, Africa

## Abstract

This study evaluated the relationship between cross-cultural social adaptation and overseas life satisfaction among Chinese medical aid team members (CMATMs) in Africa. A revised Chinese version of the Sociocultural Adaptation Scale (CSCAS) was used to measure participants’ cross-cultural social adaptation. The self-designed survey of the CMATMs’ overseas life satisfaction includes the following five aspects: food, housing, transportation, entertainment, and security. Electronic questionnaires were distributed non-randomly. Linear regression models were established to explore the association between cross-cultural social adaptation and all dimensions of overseas life satisfaction. After adjusting all the confounders, compared with moderate adaptation, poor adaptation was negatively correlated with all dimensions of overseas life satisfaction (B for food = −0.71, B for housing = −0.76, B for transportation = −0.70, B for entertainment = −0.53, B for security = −0.81, B for overall satisfaction = −0.71, all *p* < 0.001), whereas good adaptation was positively associated with all dimensions of overseas life satisfaction (B for food = 1.23, B for housing = 1.00, B for transportation = 0.84, B for entertainment = 0.84, B for security = 0.76, B for overall life satisfaction = 0.94, all *p* < 0.001). This study shows that a better cross-cultural social adaptation was positively connected to a higher level of overseas life satisfaction in general, and more specifically to higher levels of satisfaction with food, housing, transportation, entertainment, and security. This knowledge can be utilized in promoting cross-cultural social adaptation and overseas life satisfaction among CMATMs in Africa.

## 1. Introduction

With the acceleration of globalization, a greater degree of political, economic, and cultural exchanges have increasingly been occurring among different countries and regions [1]. An increasing number of individuals leave their homelands to study, work, and even settle down in other countries and regions, including international students, foreign teachers, multinational employees, and international volunteers, such as foreign medical aid teams. China, as a major party in globalization, has actively established comprehensive connections and forms of cooperation in various fields with countries and regions worldwide. In addition, as the world’s second largest economy, China has made vigorous efforts to become increasingly involved in global governance. By launching the One Belt and One Road (OBOR) project, China has shown its strong willingness to take on more responsibilities and fulfill its obligations to help other developing countries and regions, especially African countries [2,3]. China is one of the largest emerging donor countries and has provided enormous assistance toward improving the health services of more than 40 African countries [4,5,6]. It is estimated that from 2000 to 2013, China has funded 5.67 billion USD for African countries to help build their public health systems, mainly through dispatching medical aid teams [3]. Currently, there are thousands of Chinese medical aid team members (CMATMs) providing health assistance for local residents in Africa [3,5,6].

However, despite China’s active efforts toward expanding the scope of China–Africa health cooperation, CMATMs have not shown strong willingness to participate in foreign medical aid to African countries [7,8]. CMATMs can encounter multiple challenges as a result of the conditions present in many African countries, such as culture shock, and vastly different natural environments, social norms, lifestyles, and levels of economic development [9]. In transitioning from the familiar, accustomed living environment in their homeland to a foreign African country, CMATMs face an unfamiliar environment that could easily impact their psychological, mental, and somatic state. It is essential for them to quickly adapt themselves to the local natural and social environment in Africa, otherwise living with the potential triggers for disappointment and frustration that are often encountered by those faced with unfamiliar conditions could greatly influence their overseas life satisfaction. After all, it is possible that some individuals with severe problems adjusting to their new surrounding conditions could become depressed or even attempt suicide [10]. Due to the impact of the new environment, the cross-cultural social adaptation and overseas life satisfaction of the CMATMs might change accordingly, which offers a possible explanation for the low willingness to participate in foreign medical aid programs in African countries.

Life satisfaction reflects individuals’ overall cognition and self-assessment of their living conditions from a subjective psychological aspect, by comparing individuals’ living conditions with the satisfaction criteria that they hold for their lives, or at least for a certain period of their lives [11,12]. However, what is the level of satisfaction of CMATMs in African countries in terms of food, housing, transportation, entertainment, and security? What is the influencing mechanism(s) of cross-cultural adaptation and overseas life satisfaction among CMATMs, and is there a correlation? If such a correlation exists, is it positive or negative? What is the relativity? These questions have not yet been answered, since there is no correlated research or report covering relevant topics specifically targeted at CMATMs in African countries. We argue that strengthening cross-cultural adaptation can effectively improve the life quality and working situations of CMATMs in African countries. Hence, the present study thoroughly analyzed the association between the social adaptation and overseas life satisfaction of CMATMs in Africa, and provided scientific evidence for implementing relevant measures to improve the social adaptation and overseas life satisfaction of CMATMs in African countries.

## 2. Materials and Methods

### 2.1. Participants

The present study adopted a cross-sectional design. All CMATMs who were carrying out medical assistance missions in African countries during the period from 15 August 2016 to 20 November 2016 were our potential study subjects. 

### 2.2. Procedures

Our potential study subjects were all the CMATMs who were in service in African countries. The study was commissioned by the former National Health and Family Planning Commission of the People’s Republic of China (NHFPC). Prior to conducting this cross-sectional survey study, a QQ (a mobile instant messaging app with the largest user base in China.) group that included most of the CMATMs in African countries was created by the NHFPC. By using the QQ group, all of the relevant information (including the contents and procedures) of the investigation were effectively publicized in advance, and related questions raised by the CMATMs in African countries were also answered in time during the investigation. The questionnaire was designed by experts in the relevant fields, and a pilot study was previously conducted among the team leaders of the CMATMs in Africa. All electronic questionnaires were distributed non-randomly by the information platform of the Global Health Institute of Wuhan University. Three hundred and thirty-five questionnaires were returned. Out of all the questionnaires returned from the potential participants, a total of 18 questionnaires were completed with poor quality (e.g., the answers were exactly the same for all questions and/or more than a third of the questions were not answered). Therefore, these questionnaires were excluded from the final analysis. Consequently, three hundred and seventeen valid questionnaires were selected, and the effective recovery rate of the present study was 94.63%. The investigation was conducted anonymously and all the CMATMs in African countries were asked to answer the questionnaires independently. The research teams believe that potential bias should have been avoided to a certain extent.

### 2.3. Measures

The questionnaire used in this study included three sections: participants’ essential information, participants’ cross-cultural social adaptation, and participants’ overseas life satisfaction in African countries.

#### 2.3.1. Participants’ Essential Information

The first section determined the demographic characteristics of the participants, including the participants’ gender (male/female), age groups (≤40 years old, 41–50 years old, and 51–60 years old), marital status (not married/married), educational level (junior college or below, bachelor’s degree, and master’s degree or above), prior overseas experience (yes/no), type of service regions (non-hard region, class I hard region, class II hard region, class III hard region, class IV hard region, and class IV region, which indicates the toughest living conditions), gross annual income (≤130,000 CNY, 130,000–190,000 CNY, 190,000–250,000 CNY, and >250,000 CNY), and length of service (≤6 months, 6–12 months, 12–18 months, and >18 months). These indicators were considered as confounding factors in this study.

#### 2.3.2. Participants’ Cross-Cultural Social Adaptation

This section adopted the Chinese version of the Sociocultural Adaptation Scale (CSCAS), as developed by Qi and Li [13]. The CSCAS has been widely used in many studies on cross-cultural social adaptation among international students in China. For the targeted participants of this study, we fine-tuned a few items of the original scale (instances of the term “campus” in the original items were replaced by “African countries”, e.g., (a) understanding of African culture and Africans; (b) adapt to the African climate; (c) adapt to the pace of life in African countries). The CSCAS contains 20 questions, and all questions use a 5-point Likert scale from very poor cross-cultural social adaptation (scored 1 point) to very good adaptation (scored 5 points). The total score of the CSCAS is between 20–100 points, and a total score of 40 or less is considered poor adaptation, 41 to 79 points is considered moderate adaptation, 80 or above is considered good adaptation. The Cronbach’s alpha coefficient of the CSCAS was 0.93 in the present study.

#### 2.3.3. Participants’ Overseas Life Satisfaction in African Countries

The third section was the self-assessment questionnaire of the CMATMs’ overseas life satisfaction in African countries. This self-designed measurement tool included a total of 18 questions in five aspects, including food, housing, transportation, entertainment, and security. The questionnaire also adopted a 5-point Likert scale, and the various aspects and the overall average scores were calculated. A higher score indicated a higher level of overseas life satisfaction in African countries. In this study, the Cronbach’s alpha coefficient of the questionnaire was 0.91.

### 2.4. Statistical Analysis

The Statistical Package for the Social Sciences (SPSS) version 22.0 for Windows (SPSS Inc., Chicago, IL, USA) was used to perform all the data analyses, with a two-sided *p*-value of less than 5% for all tests indicating statistically significant differences. The content of the analysis was composed of three main parts. First, the overall characteristics of the study population as well as the characteristics of the study population by level of cross-cultural adaptation were initially summarized by descriptive analysis. Chi-square tests were used to examine the differences in demographic characteristics among the CMATMs with different levels of cross-cultural adaptation (Table 1). The average scores of all dimensions of overseas life satisfaction are presented as mean and standard deviation. Analysis of variance and the Student–Newman–Keuls method were used to identify differences in all dimensions of overseas life satisfaction between different cross-cultural social adaptation groups (Table 2). Finally, linear regression models were established to explore the association between cross-cultural social adaptation and all dimensions of overseas life satisfaction among CMATMs in African countries. Both the unadjusted models and adjusted models are reported in Table 3. The results are presented as an unstandardized coefficient (B) and with a 95% confidence interval (95% CI).

### 2.5. Ethical Statements

The participants were guaranteed no risk involved in participating in the survey, and the survey was conducted anonymously online. Participants were assured that the data would be treated confidentially. Our study was conducted in accordance with the Declaration of Helsinki, and the study protocol was reviewed and approved by the institutional review board of School of Health Science of Wuhan University (Project Identification Code: 2016-S-0011-4/03).

## 3. Results

### 3.1. Demographic Characteristics of the Study Sample

A total of 317 individuals composed of 109 (34.38%) women and 208 (65.62%) men were involved in the cross-sectional study, including 108 (34.07%) respondents with poor cross-cultural adaptation, 148 (46.69%) with moderate adaptation, and 61 (19.24%) with good adaptation. Most of the participants were in their forties (46.37%) and married (85.80%). Of the respondents, 63.09% received undergraduate education and 72.87% had no prior overseas experience. Slightly more than a quarter (27.44%) of participants worked in class II hard regions and nearly one-third (32.18%) earned between 130,000 and 190,000 CNY a year. About 40% participants had provided service for 6–12 months. Significant differences were revealed on the distribution of age (χ^2^ = 13.724, *p* = 0.008), prior overseas experience (χ^2^ = 7.549, *p* = 0.023), and the length of service (χ^2^ = 18.265, *p* = 0.006) among the CMATMs with different levels of cross-cultural adaptation (Table 1).

### 3.2. Life Satisfaction Scores of the Study Population

Life satisfaction was measured from five dimensions, including food, housing, transportation, entertainment, and security. The overall life satisfaction scores of participants with poor adaptation, moderate adaptation, and good adaptation were 2.15 ± 0.49, 2.90 ± 0.36, and 3.82 ± 0.51, respectively. The highest scores were observed in transportation, while the lowest scores were found in entertainment, regardless of the level of adaptation. The mean values of life satisfaction scores were significantly different in all dimensions and the overall level among individuals with different levels of adaptation (*p* < 0.001). Furthermore, the life satisfaction scores increased with increases in the degree of cross-cultural adaptation across all dimensions (Table 2).

### 3.3. Association between Cross-Cultural Adaptation and Life Satisfaction Scores

As demonstrated in Table 3, the results of the linear regression analysis were consistent in all models. The final parsimonious model (model 5) revealed that, compared with moderate adaptation, poor adaption was negatively correlated with the life satisfaction of food (B = −0.71, CI = −0.88, −0.54, *p* < 0.001), housing (B = −0.76, CI = −0.94, −0.57, *p* < 0.001), transportation (B = −0.70, CI = −0.89, −0.50, *p* < 0.001), entertainment (B = −0.53, CI = −0.70, −0.36, *p* < 0.001), security (B = −0.81, CI = −1.04, −0.58, *p* < 0.001), and the overall satisfaction (B = −0.71, CI = −0.82, −0.60, *p* < 0.001), whereas good adaptation was positively associated with the life satisfaction of food (B = 1.23, CI = 1.03, 1.44, *p* < 0.001), housing (B = 1.00, CI = 0.77, 1.22, *p* < 0.001), transportation (B = 0.84, CI = 0.60, 1.08, *p* < 0.001), entertainment (B = 0.84, CI = 0.64, 1.05, *p* < 0.001), security (B = 0.76, CI = 0.49, 1.04, *p* < 0.001), and the overall life satisfaction (B = 0.94, CI = 0.81, 1.07, *p* < 0.001).

## 4. Discussion

From the perspective of global governance, providing medical assistance to African countries is a typical manifestation of China’s provision of public goods for the global health governance system [3]. Furthermore, dispatching CMATMs is one of the most effective ways to help solve the health dilemma in African countries and to improve their health situation and safeguard the health and safety of local residents [4,5,6]. Owing to the differences in political, economic, sociocultural, and religious beliefs between China and African countries, CMATMs might have to change their ideology, behaviors, and lifestyles in response to the new social environment. Their adaptability to implement these changes will inevitably affect their overseas life satisfaction [7,8,9]. Therefore, this study focused on the association between cross-cultural social adaptation and life satisfaction among CMATMs in Africa. The results of the study will provide insights into proposals and the implementation of the various relevant coping strategies for CMATMs in Africa. 

China is playing an increasingly essential role in the development of health assistance in other countries. Undoubtedly, the importance of medical teams—one of the primary ways in which China helps provide medical aid to Africa—cannot be trivialized [14]. Previous studies have pointed out that there was a significant association between working conditions and job performance/satisfaction [15,16,17]. Additionally, Tucker et al. found that job efficiency was significantly associated with cross-cultural adaptation among corporate expatriates [18]. However, to the best of our knowledge, there have been few research studies about life satisfaction and intercultural adjustment of the personnel in medical aid teams. Furthermore, life satisfaction has been seldom measured in terms of food, housing, transportation, entertainment, and security. 

In this study, we examined the correlation between intercultural adjustment and the life satisfaction of workers in foreign aid medical teams. The results revealed positive relationships between cross-cultural social adaptation and each dimension of life satisfaction, and the association still remained the same after adjustment for all confounders. A longitudinal analysis conducted in Portuguese college students indicated that higher academic adaptation may lead to better global life satisfaction [19]. Moreover, as Joardar and Weisang pointed out, an inability to adapt may result in dissatisfaction [20]. Another study suggested that the adjustment to and satisfaction with retirement were correlated [21]. Although these studies focused on different populations, the results were actually similar and in line with our expectations. 

In the life satisfaction assessment, transportation scored the highest, whereas entertainment scored the lowest regardless of the degree of adaptation. There are several explanations for this phenomenon. As indicated by Davies et al., a large number of concessional loans provided by The Export-Import Bank of China were invested in infrastructure development [22]. Additionally, Foster et al. further pointed out that transport, especially rail and road, was one of the sectors that benefited the most [23]. With the implement of “The Silk Road Economic Belt and 21st Century Maritime Silk Road”, more attention was attached to transportation for better interconnectivity between China and related countries [24]. Therefore, in many areas the traffic conditions have been greatly improved, which may help to make workers in the medical teams satisfied. Nevertheless, the entertainment situation was not so promising, which may be explained by the lack of leisure time due to high-intensity work, limited social circles, and monotonous activities. Magdalena Bjerneld et al. conducted interviews with Swedish medical personnel who had participated in assistance for health [25]. Their study found that tough assignments and heavy burdens were stressors for respondents, and some participants felt isolated because of language barriers [25]. Therefore, our study suggests that more attention should be paid to the recreational life of the staff. The situation may be changed by reducing their work burden, providing training for non-medical knowledge, and enriching forms of activities.

This study also found that security was positively correlated with cross-cultural adaptation. Insecurity may be caused by the danger of infectious diseases and terrorist attacks. For one thing, the destinations of foreign aid are usually developing countries with high prevalence of infectious diseases, such as malaria, Dengue, and Ebola. As mentioned in a previous article, morbidity was relatively high among travelers who visited developing countries, and the type of disease was correlated with travel destinations [26]. In terms of other security-related considerations, according to Aid Worker Security Report 2018, 313 aid workers were involved in 158 major incidents of violence, in which 139 were killed, 102 were wounded, and 72 were kidnapped in 2017 [27]. Furthermore, a previous study regarding the reasons of those negative events revealed that unstable political circumstances, underdeveloped economies, and the existence of armed conflicts would increase the risk of violent attacks [28]. Hence, action must be taken to promote a sense of safety so that members of medical aid teams can better adapt to the new environments.

Associations of intercultural adjustment with satisfaction of food and housing were revealed in our study as well. The results were similar to the findings of Schure et al., who showed that food and housing insecurity may be detrimental to physical and psychological health [29]. As a matter of fact, shortages of food are still a problem that can bother medical aid workers and some medical aid workers have reported losing weight over the duration of their foreign aid [25]. It is understandable that the considerable differences in eating habits may have an impact on international adaptation. Accommodation of medical aid workers is provided by the recipient country. Poor living conditions may include crowded living conditions and noisy environments [29]. The relevant reports of housing are limited, which indicates a current lack of attention on the matter. Therefore, improving the food and housing conditions could be an effective way to promote the life satisfaction and cross-cultural adjustment of foreign medical aid workers.

A number of limitations of this study should be elucidated. For instance, some confounders, especially those that are more related to subjective well-being and other variables more related to psycho-social aspects, were not included as covariates in the data analyses. Secondly, this study may have provided a general overview of the association between cross-cultural social adaptation and overseas life satisfaction among CMATMs in Africa, yet there are still a lot of knowledge gaps that need to be filled (for example, whether this kind of relationship will change at different times). For future studies, it is necessary to explore this issue from different research perspectives in combination with a multidisciplinary approach in order to allow for a more in-depth study.

## 5. Conclusions

In conclusion, our study showed that overseas life satisfaction in general, and also more specifically in terms of food, housing, transportation, entertainment, and security, was positively related to cross-cultural social adaptation among Chinese medical aid team members (CMATMs) in Africa. The present study indicated the possibility that improving the overseas living conditions, especially the food, housing, transportation, entertainment, and security conditions, could be an effective way to promote cross-cultural social adaptation for CMATMs in Africa. 

## Figures and Tables

**Table 1 ijerph-16-01572-t001:** Characteristics of the study population by level of cross-cultural adaptation (*n* = 317).

Variables	All Participants		Poor Adaptation		Moderate Adaptation		Good Adaptation	
	*n*	%	*n*	%	*n*	%	*n*	%
**Gender**	χ^2^ = 1.2800, *p* = 527							
Female	109	34.38	41	37.96	50	33.78	18	29.51
Male	208	65.62	67	62.04	98	66.22	43	70.49
**Age**	χ^2^ = 13.724, *p* = 0.008							
≤40	98	30.91	33	30.56	48	32.43	17	27.87
41–50	147	46.37	51	47.22	76	51.35	20	32.79
51–60	72	22.71	24	22.22	24	16.22	24	39.34
**Marital status**	χ^2^ = 5.685, *p* = 0.058							
Not married	45	14.20	22	20.37	18	12.16	5	8.20
Married	272	85.80	86	79.63	130	87.84	56	91.80
**Education level**	χ^2^ = 4.711, *p* = 0.318							
Junior college or below	34	10.73	10	9.26	13	8.78	11	18.03
Bachelor’s degree	200	63.09	70	64.81	93	62.84	37	60.66
Master’s degree or above	83	26.18	28	25.93	42	28.38	13	21.31
**Prior overseas experience**	χ^2^ = 7.549, *p* = 0.023							
No	231	72.87	69	63.89	112	75.68	50	81.97
Yes	86	27.13	39	63.11	36	24.32	11	18.03
**Type of service regions**	χ^2^ = 11.766, *p* = 0.162							
Non-hard region	62	19.56	17	15.74	29	19.59	16	26.23
Class I hard region	79	24.92	29	26.85	42	28.38	8	13.11
Class II hard region	87	27.44	30	27.78	42	28.38	15	24.59
Class III hard region	44	13.88	13	12.04	21	14.19	10	16.39
Class IV hard region	45	14.20	19	17.59	14	9.46	12	19.67
**Gross annual income**	χ^2^ = 7.262, *p* = 0.297							
≤130,000	61	19.24	20	18.52	33	22.30	8	13.11
130,000–190,000	102	32.18	32	29.63	53	35.81	17	27.87
190,000–250,000	97	30.60	38	35.19	36	24.32	23	37.70
>250,000	57	17.98	18	16.67	26	19.57	13	21.31
**Length of service**	χ^2^ = 18.265, *p* = 0.006							
≤6 months	70	22.08	21	19.44	42	28.38	7	11.48
6–12 months	34	10.73	14	12.96	12	8.11	8	13.11
12–18 months	124	39.12	32	29.63	62	41.89	30	49.18
>18 months	89	28.08	41	37.96	32	21.62	16	26.23
**Overall**	317	100.00	108	34.07	148	46.69	61	19.24

**Table 2 ijerph-16-01572-t002:** Life satisfaction scores of the study population by cross-cultural adaptation (data presented as mean ± standard deviation).

Dimensions of Life Satisfaction	Poor Adaptation	Moderate Adaptation	Good Adaptation	F	*p*	Pairwise Comparison
Food	1.98 ± 0.70	2.78 ± 0.70	3.87 ± 0.75	139.685	<0.001	1 < 2 < 3
Housing	2.31 ± 0.81	3.14 ± 0.77	4.11 ± 0.74	106.593	<0.001	1 < 2 < 3
Transportation	2.70 ± 0.81	3.45 ± 0.72	4.25 ± 0.91	77.495	<0.001	1 < 2 < 3
Entertainment	1.66 ± 0.63	2.20 ± 0.65	3.09 ± 0.86	84.240	<0.001	1 < 2 < 3
Security	2.19 ± 1.00	2.99 ± 0.88	3.73 ± 0.78	59.901	<0.001	1 < 2 < 3
Overall	2.15 ± 0.49	2.90 ± 0.36	3.82 ± 0.51	291.472	<0.001	1 < 2 < 3

**Table 3 ijerph-16-01572-t003:** Linear regression models testing the association between cross-cultural adaptation and life satisfaction scores.

Model	Food		Housing		Transportation		Entertainment		Security		Total	
	*B*	(95% CI)	*B*	(95% CI)	*B*	(95% CI)	*B*	(95% CI)	*B*	(95% CI)	*B*	(95% CI)
**Model 1**												
(Constant)	2.78	(2.67, 2.90) ***	3.14	(3.01, 3.27) ***	3.45	(3.32, 3.58) ***	2.20	(2.09, 2.31) ***	2.99	(2.84, 3.14) ***	2.89	(2.83, 2.97) ***
Moderate adaptation (Control group)												
Poor adaptation	−0.80	(−0.98, −0.63) ***	−0.83	(−1.03, −0.64) ***	−0.75	(−0.94, −0.55) ***	−0.54	(−0.71, −0.37) ***	−0.80	(−1.03, −0.57) ***	−0.74	(−0.85, −0.63) ***
Good adaptation	1.09	(0.88, 1.30) ***	0.98	(0.74, 1.21) ***	0.79	(0.56, 1.03) ***	0.89	(0.68, 1.10) ***	0.74	(0.47, 1.01) ***	0.92	(0.79, 1.05) ***
**Model 2**												
(Constant)	2.79	(2.41, 2.72) ***	3.07	(2.66, 3.49) ***	3.55	(2.81, 3.41) ***	2.20	(1.84, 2.57) ***	2.67	(2.20, 3.15) ***	2.86	(2.64, 3.10) ***
Moderate adaptation (Control group)												
Poor adaptation	−0.79	(−0.96, −0.61) ***	−0.83	(−1.03, −0.63) ***	−0.75	(−0.95, −0.56) ***	−0.54	(−0.71, −0.36) ***	−0.77	(−0.99, −0.55) ***	−0.74	(−0.85, −0.63) ***
Good adaptation	1.11	(0.90, 1.33) ***	0.97	(0.73, 1.22) ***	0.76	(0.52, 1.00) ***	0.92	(0.71, 1.14) ***	0.77	(0.49, 1.04) ***	0.93	(0.80, 1.07) ***
**Model 3**												
(Constant)	3.33	(3.03, 3.62) ***	3.38	(3.06, 3.70) ***	3.51	(3.17, 3.85) ***	2.27	(1.98, 2.57) ***	2.97	(2.58, 3.36) ***	3.11	(2.93, 3.30) ***
Moderate adaptation (Control group)												
Poor adaptation	−0.71	(−0.89, −0.54) ***	−0.75	(−0.93, −0.56) ***	−0.73	(−0.93, −0.53) ***	−0.55	(−0.73, −0.38) ***	−0.82	(−1.05, −0.59) ***	−0.71	(−0.82, −0.60) ***
Good adaptation	1.20	(0.99, 1.40) ***	0.99	(0.77, 1.22) ***	0.83	(0.59, 1.07) ***	0.84	(0.63, 1.05) ***	0.70	(0.43, 0.98) ***	0.95	(0.82, 1.08) ***
**Model 4**												
(Constant)	3.27	(2.81, 3.72) ***	3.36	(2.59, 3.85) ***	3.53	(2.99, 4.06) ***	2.24	(1.78, 2.70) ***	2.60	(2.00, 3.20) ***	3.04	(2.75, 3.33) ***
Moderate adaptation (Control group)												
Poor adaptation	−0.70	(−0.87, −0.52) ***	−0.76	(−0.95, −0.57) ***	−0.73	(−0.94, −0.52) ***	−0.54	(−0.72, −0.37) ***	−0.78	(−1.01, −0.55) ***	−0.69	(−0.81, −0.59) ***
Good adaptation	1.21	(1.00, 1.42) ***	0.96	(0.73, 1.19) ***	0.80	(0.56, 1.05) ***	0.88	(0.67, 1.10) ***	0.75	(0.48, 1.03) ***	0.95	(0.82, 1.09) ***
**Model 5**												
(Constant)	3.33	(3.10, 3.56) ***	3.37	(3.05, 3.68) ***	3.63	(3.43, 3.82) ***	2.26	(2.00, 2.51) ***	2.80	(2.43, 3.18) ***	3.10	(2.93, 3.26) ***
Moderate adaptation (Control group)												
Poor adaptation	−0.71	(−0.88, −0.54) ***	−0.76	(−0.94, −0.57) ***	−0.70	(−0.89, −0.50) ***	−0.53	(−0.70, −0.36) ***	−0.81	(−1.04, −0.58) ***	−0.71	(−0.82, −0.60) ***
Good adaptation	1.23	(1.03, 1.44) ***	1.00	(0.77, 1.22) ***	0.84	(0.60, 1.08) ***	0.84	(0.64, 1.05) ***	0.76	(0.49, 1.04) ***	0.94	(0.81, 1.07) ***

Note: B = Coefficient; “*” *p*-value < 0.05; “**” *p*-value < 0.01; “***” *p*-value < 0.001. Model 1 is the unadjusted model. Model 2 adjusted for gender, age, marital status, and education level. Model 3 adjusted for prior overseas experience, type of service regions, gross annual income, and length of service. Model 4 adjusted for gender, age, marital status, education level, prior overseas experience, type of service regions, gross annual income, and length of service. Model 5 is the final parsimonious model, adjusted for predictors that were significantly associated with each of the dimensions of life satisfaction and the overall life satisfaction in Model 4. Specifically, age, gross annual income, and length of service were significantly associated with participants’ satisfaction of food; type of service regions, gross annual income, and length of service were significantly associated with participants’ satisfaction of housing; length of service was significantly associated with participants’ satisfaction of transportation; type of service regions and length of service were significantly associated with participants’ satisfaction of entertainment; type of service regions, age, marital status, and length of service were significantly associated with participants’ satisfaction of security; type of service regions, overseas experience, and length of service were significantly associated with participants’ overall life satisfaction.

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
