# Peer review of "Association between Cross-Cultural Social Adaptation and Overseas Life Satisfaction among Chinese Medical Aid Team Members (CMATMs) in Africa"

_ijerph, 2019, doi:10.3390/ijerph16091572_

Round 1
Reviewer 1 Report
Dear authors,
Thank you for submitting "Association between cross-cultural social adaptation and overseas life satisfaction among Chinese medical aid team members (CMATMs) in Africa" for International Journal of Environmental Research and Public Health.
This study is about the importance of adaptation of medical aid workers and the relationship of adaptation with satisfaction of five different domains. The importance of this topic is well-raised in the manuscript, it convinced me as a reader that this topic is highly relevant and has important implications.
Although the importance is clear, some other aspects of the manuscript and the study are weak and can be improved:
Major issues:
- I am not convinced that the methods are the most appropriate for the issue you want to study. Some changes you may want to consider:
a) Theoretical substantiation lacks; you may want to add a theoretical framework on adaptation; it will provide more arguments for the expectations and statements.
b) A conceptual framework and related hypotheses can be added and guide the reader through the study, the link with methods would be more clear.
c) Please try other methods; for example: cluster analysis, structural equation models, person centered approach, etc. Also, the presentation of the results might be adapted: for example: the percentages over the different groups of adaptation (for example: table 1; of the non-married group; how many people rate low adapted, etc.). And also, can you explain why the adaptation groups are seperated in three groups and not more? Why do you use groups instead of original scores? It takes away variation and thus explained variance. Please also add the explained variance.
- The writing is very concise; this can be a strength, but at this stage, the reader lacks information and needs some more information. The manuscript needs elaboration on various aspects (theoretical fundaments, methods, ...).
Small issues:
- The abstract is rather good; depending on the analyses, you may want to strengthen your results (for example: "correlated" is less convincing than "regressions").
- Each statement needs substantiation with references (for example: page 2, line 42).
- Some substantiation is currently in the discussion part; for example: line 233-238, line 226, these parts would be more appropriate in the introduction or theoretical argumentations (before methods).
- Provide more structure in your writing, signalling words. For example: "we have three reasons why ... First, .... Second, ... etc.".
- What is a QQ group?
- What do the regions mean?
- Provide sample items.
- Please show all items for new scales.
- Does +- mean standard deviation?
- What do you mean by "non-randomly"?
- Can you elaborate on the extent to which individuals had experience overseas? You may want to consider to take this explicit into account in the analyses.
- Please mention the covariates in the table (and, thus, also the results for these parameters).
In brief: it is a highly relevant topic and study, but needs some more information in order to strengthen the study and its analyses.
Good luck with this important study!
Author Response
April 21, 2019
Editor-in-Chief
Int. J. Environ. Res. Public Health
Dear Editors and Reviewers,
We are respectfully submitting a revised version of our manuscript, entitled “Association between cross-cultural social adaptation and overseas life satisfaction among Chinese medical aid team members (CMATMs) in Africa”, for your consideration for publication in the International Journal of Environmental Research and Public Health.
The authors very much appreciate the thoughtful and critical feedback from the reviewers. We are delighted at your decision. The reviewers are clearly familiar with the topic, as is exemplified by their close and accurate review of this manuscript. The manuscript has been revised according to the reviewers’ comments, and all changes have been highlighted for ready identification. In addition, we have addressed each of the comments from you specifically, and our responses have been outlined in a comment/response format below (see responses to the comments).
Hope the revision is satisfactory and this manuscript is now acceptable for publication in your journal.
I am looking forward to hearing from you soon.
Sincerely,
Zongfu Mao, Ph.D.
Professor and Director
Global Health Institute
School of Health Science
Wuhan University
115# Donghu Road, Wuhan City 430071,P.R.China
E-mail: zfmao@whu.edu.cn

Reviewer 2 Report
In general, the study presented shows optimal conditions for its publication
The reading of the contents is clear and concise, clarifying the concepts necessary for understanding it.
The theme is innovative, the methodology is adequate and the discussion associates previous references on the concepts that arise in the objectives of the study
For the improvement of this investigation and deepening it is later publications, it is recommended to study satisfaction from a perspective more related to subjective well-being and other variables more related to psychosocial aspects
On the other hand, it would be convenient for the authors to specify the limitations of the study, as well as possible future lines of research on this subject
Author Response
April 21, 2019
Editor-in-Chief
Int. J. Environ. Res. Public Health
Dear Editors and Reviewers,
We are respectfully submitting a revised version of our manuscript, entitled “Association between cross-cultural social adaptation and overseas life satisfaction among Chinese medical aid team members (CMATMs) in Africa”, for your consideration for publication in the International Journal of Environmental Research and Public Health.
The authors very much appreciate the thoughtful and critical feedback from the reviewers. We are delighted at your decision. The reviewers are clearly familiar with the topic, as is exemplified by their close and accurate review of this manuscript. The manuscript has been revised according to the reviewers’ comments, and all changes have been highlighted for ready identification. In addition, we have addressed each of the comments from you specifically, and our responses have been outlined in a comment/response format below (see responses to the comments).
Hope the revision is satisfactory and this manuscript is now acceptable for publication in your journal.
I am looking forward to hearing from you soon.
Sincerely,
Zongfu Mao, Ph.D.
Professor and Director
Global Health Institute
School of Health Science
Wuhan University
115# Donghu Road, Wuhan City 430071,P.R.China
E-mail: zfmao@whu.edu.cn
Responses to the reviewers’ comments
COMMENTS
In general, the study presented shows optimal conditions for its publication.
The reading of the contents is clear and concise, clarifying the concepts necessary for understanding it.
The theme is innovative, the methodology is adequate and the discussion associates previous references on the concepts that arise in the objectives of the study.
Response: We thank the reviewer for the supportive review comments and encouragement.
For the improvement of this investigation and deepening it is later publications, it is recommended to study satisfaction from a perspective more related to subjective well-being and other variables more related to psychosocial aspects.
On the other hand, it would be convenient for the authors to specify the limitations of the study, as well as possible future lines of research on this subject.
Response: We thank the reviewer for pointing out this deficiency and we have added the related information in the revised manuscript as suggested accordingly. Please see the last paragraph of the discussion.